# Assessing Expressive Oral Reading Fluency

**Timothy G. Morrison \*** and **Brad Wilcox**

Department of Teacher Education, Brigham Young University, Provo, UT 84602, USA; brad_wilcox@byu.edu
* Correspondence: tim_morrison@byu.edu

**Abstract:** Educators struggle to assess various aspects of reading in valid and reliable ways. Whether it is comprehension, phonological awareness, vocabulary, or phonics, determining appropriate assessments is challenging across grade levels and student abilities. Also challenging is measuring aspects of fluency: rate, accuracy, and prosody. This article presents a history of fluency in American education with particular focus on assessing expressive oral reading. In addition, the two major approaches to prosody assessment will be explained, and the three most prominent tools for rating expressive oral reading will be analyzed and discussed.

**Keywords:** fluency; prosody; NAEP; MDFS; spectrographic measurement

## 1. History

From the colonial period in America until the early 1900s, one major focus of beginning literacy instruction was expressive oral reading. During the 1600s through the end of the 1700s, "the aim of developing eloquent oral reading was paramount..." [1]. Children were expected to read texts—especially the Bible—expressively. The limited number of suggestions and instructions provided to teachers during this period encouraged use of "exercises for pronunciation and enunciation" [1,2]. Although the types of selections and methods of instruction varied through the mid-1800s, expressive oral reading continued to be stressed in U.S. schools.

A major transition from stressing oral reading to attending to silent reading occurred in the early 1900s, initially led by Francis W. Parker [1]. He distinguished between expression (speech and oral reading) and attention (silent reading). Parker and others argued that silent reading should be preferred over oral reading, because most of the reading that adults do in their lives is silent, and this form of reading leads to greater rate and increased comprehension. This movement toward privileging silent over oral reading was supported by William S. Gray, major author of the popular Dick and Jane books and the preeminent reading figure of his day. This change in focus shifted from performing text to comprehending it [2].

For decades, the attention paid to silent reading and comprehension diverted attention from fluency, including expressive oral reading. The prevailing view then was that once readers had identified the words in text, they would understand what they had read through use of oral language comprehension processes [3]. Reading educators and researchers discussed most effective approaches to reading instruction, focusing on whole word vs. phonics instruction. The ensuing reading wars continued for decades, and fluency instruction was largely overlooked in classroom practice.

In the 1960s and 1970s various models of the reading process were developed, including one in which automaticity of word identification was a key element [4]. As theoreticians began to attend to the necessity of readers to automatically identify words quickly and accurately, some called for teachers to encourage the practice of oral reading fluency with their students. The method of repeated reading [5] and Neurological Impress [6] were developed to help readers increase word identification abilities of students so that they could use more cognitive resources to attend to meaning.

It was in this context that Allington published his article, "Fluency: The Neglected Goal" [7], in which he highlighted the need to return attention to fluency in elementary school instruction. He encouraged attention to developing readers' rates and accuracy of oral reading, but also reminded teachers to focus on expressive reading. Allington was not alone in reminding teachers of the need to focus on developing fluent readers. Collins compared the prosody of good and poor readers at a time when most of the focus on fluency instruction was rate and accuracy. Collins reported that teachers often focused struggling readers' attention on saying the words correctly, while proficient readers were encouraged to make the text sound right when they read orally [8].

Even though this research attended to expressive oral reading, the inclusion of prosody in a definition of fluency was not cemented in many people's minds until the Report of the National Reading Panel in 2000 [9]. With the publication of that pivotal report, most in the literacy research and teaching communities included three aspects of fluent oral reading—rate, accuracy, and expression. However, the ease of assessing rate and accuracy resulted in an emphasis on those two factors over prosody.

## 2. Prosody

Prosody refers to elements of speech that go beyond ability to produce vowel and consonant sounds to the ability to use appropriate intonation, tone, stress, and rhythm when reading connected text. When young children are first learning, they read orally, not silently. Caregivers and others read orally to preschoolers, and their first attempts at reading are oral. As beginners, many children read with very little expression when they try to verbalize the words they see in text. As their word identification becomes more automatic, they increase their rate of reading, allowing them to allocate cognitive resources to comprehension and oral reading expression [10].

Many teachers and others prompt children to make their oral reading sound like they are talking, not reading with a monotone voice. "When a child reads prosodically, oral reading sounds much like speech with appropriate phrasing, pause structures, stress, rise and fall patterns, and general expressiveness" [11].

Prosody seems to be closely associated with reading comprehension. Even though researchers discuss fluency differently, most agree that the link between fluency and students' understanding of the text they encounter is substantial. Those who read with good expression also tend to have better reading comprehension abilities than can be explained by reading rates alone [12]. They recognize the phrasal and syntactical structures the author is using and are able to deliver the text orally as the author intended. However, this alone does not guarantee comprehension. It appears that oral reading fluency can sometimes limit or support comprehension [13]. When children read very slowly and with poor accuracy, their comprehension can suffer, even if they read expressively. As children read faster and with greater accuracy, their reading comprehension can also increase, even if their prosody does not improve. Thus, as children increase their rate, accuracy, and expression, a question arises: Does improved prosody lead to greater reading comprehension, or does comprehension of text lead to the ability to read with greater expression?

Although the research is not conclusive, it seems to indicate that there are differential implications for both proficient and struggling readers. "Some researchers have suggested that becoming a fluent reader has more to do with focusing on meaning construction than it has to do with attending to the words on a page" [14]. As proficient readers attend to meaning during reading, their oral reading expression demonstrates their understanding. But expressive oral reading influences struggling readers in a different way. Strugglers are typically more focused on word identification. As they work to identify words, the flow of the text is interrupted and their lack of expressive reading demonstrates their poor understanding [15]. The best way to characterize the research on prosody and reading comprehension may be summed up by Schwanenflugel and Kuhn: "It has been argued that the relationship between fluency and comprehension is bidirectional: both reciprocal and interactive" [15].

## 3. Approaches to Prosody Assessment

Teachers provide instruction and model oral reading to help children develop their own reading fluency abilities. In addition, they need to assess children's fluency development, including expressive oral reading. Assessing oral reading rate and accuracy is relatively straightforward. Teachers and researchers can easily measure students' reading rates by establishing the number of words students read per minute (wpm). They can also measure the number of words students read correctly as they read (wcpm). The 2002 NAEP assessment of fluency for fourth graders showed that students read at an average rate of 119 words per minute, and that 90% of students read at least 95% of words accurately, when scoring only meaning-change miscues as errors. As rate of reading increased, comprehension scores also increased. Measuring effective prosodic reading, however, is more challenging.

Two general approaches to assessing expressive oral reading have been developed. One approach does not depend on rater judgments of students' reading, opting instead for a more objective method. Recordings of oral reading are analyzed by automated means to measure elements of speech, such as pitch, intensity, and duration [16,17]. In a second approach, human raters make subjective judgments about individuals' prosodic reading utilizing rubrics that describe various aspects of expressive oral reading [14,18].

## 4. Methodology

As we reviewed the literature on both of these approaches, we employed the following methodology. We first identified leading search engines used for academic literature reviews: Google Scholar (scholar.google.com); ResearchGate (researchgate.net); and VirtualLRC (cse.google.com). We then used the following search terms to locate relevant articles: assessing expressive oral reading; NAEP fluency 2002; Multidimesional Fluency Scale; Comprehensive Oral Reading Fluency Scale (CORFS).

We focused on articles that appeared in the results of searches conducted in at least two of the search engines. We then focused on studies that appeared consistently in the reference lists of these articles. We also paid close attention to studies that had been conducted in international settings to indicate the widespread use of these assessment measures.

## 5. Automated Assessment

One method to gauge an individual's expressive oral reading is to examine recordings of specific features of speech [11]. After recordings of an individual's reading have been made, the contours of their speech are analyzed using computer software programs that depict them visually. The software that is commonly used for this is Praat, developed in 2001 by Dutch researchers, Paul Boersma and David Weenink [19,20]. It has undergone several revisions and remains the most important tool for analyzing speech because it is easily available and user friendly.

Praat has been used by linguists in the field of phonetics to study specific features of speech to understand the sound patterns of normal English [21]. The Praat software has also been used to teach those learning English as a foreign language (EFL) to understand prosodic features of English [22]. As individuals become more aware of the sounds of English, they can practice in more focused ways and use the language in ways that sound more like native English speakers. In addition, Praat software has been used to assist those who have been affected by vocal cord paralysis. By examining their speech patterns as they try to improve their speech production, patients can learn to more closely match the pitch and pause patterns in English [23].

The most common automated assessment in education settings provides spectrographic measurements of speech. Analyses of the graphic displays of oral reading can highlight specific elements. The two most common aspects of oral reading that have been examined using this software are pauses and pitch [24]. Various elements of both have been measured using Praat software to provide greater understanding of readers' prosodic practices.

When analyzing pauses in recorded oral reading, examiners consider the ratio of actual and grammatically-expected pauses within sentences. This analysis can determine if a reader's pauses are expected and appropriate, or if they tend to be ungrammatical and indicate unjustified pausing practices [24]. The more closely a reader's actual number of pauses match those expected by the grammar of the text, the more reliably researchers can judge whether or not the child is reading with appropriate phrasing.

When evaluating pitch in oral reading, examiners consider how readers raise and drop their voices. Effective readers are more emphatic in pitch variation than struggling readers [25]. The magnitude of the decrease in pitch during reading is measured to determine whether the declination is appropriate. Measuring pitch of oral reading also examines the general up and down pitch swings in a reader's voice. These variations in pitch are generally considered to be equated with appropriate expressive reading. When such variation of pitch is not present in a child's oral reading, the reading usually comes across as flat and monotone.

Along with pausing and pitch, stress is another property of prosody. However, stress is difficult to isolate and measure because it includes broader concepts, including pitch, duration, and intensity [26–28]. When teachers focus attention on pitch, issues related to stress will generally improve as well [29].

Nationally and internationally, researchers have used automated assessment of prosody. Researchers in Spain [30] asked 103 third- through sixth-grade students to orally read four expository texts and answer comprehension questions. Using Praat software, they measured typical aspects of prosody and found that children with lower levels of reading comprehension made more inappropriate pauses and unacceptable levels and durations of pitch compared to more able readers.

Ardoin et al. examined the role of repeated readings and wide reading in improving multiple dimensions of reading, including fluency. They asked 168 second graders to practice reading four times each week over a nine-week period. Using the Praat software, they found that both repeated reading and wide reading were effective in improving reading fluency, which in turn affected other reading behaviors, including expressive oral reading. The second graders in this study improved in both pitch and pause scores [31].

Researchers have examined pitch and pause durations and changes during oral reading to measure prosodic reading of adult readers. They examined these aspects of expressive reading in relation to adult readers' scores on tests of decoding, word identification, and comprehension. For those with limited reading skills, patterns of pausing accounted for a significant amount of variance in comprehension scores [32].

## 6. Human Assessment

In addition to automated, spectrographic measurements of expressive oral reading, educators can instead use rating scales to judge quality of prosody. Rubrics establish criteria for human judgment of acceptable performance of specific tasks. They are commonly used to in classroom settings to systematically evaluate student's abilities and behaviors, especially with processes that are not easily measured in other ways. A performance can be designed to measure a student's ability, knowledge, and skills. For example, a student may be asked to demonstrate some physical or artistic achievement, play a musical instrument, create or critique a work of art, or improvise a dance or a scene. These kinds of performances, tasks, projects, and portfolios can be scored using rubrics. Rubrics allow researchers and teachers to clarify components of a skill and permit them to make judgments about what students know and can do in relation to specific objectives. Observers can use rubrics to judge the degree, frequency, or range of student behaviors and understand the degree to which a student has mastered a skill [33].

Some rubrics provide for holistic evaluation. Using a global approach, a set of interrelated tasks is identified that contributes to the whole. Using this type of rubric, a teacher or researcher can evaluate

quickly and efficiently to provide an overall impression of ability. However, holistic evaluations do not provide the detail available in analytic approaches.

Analytic rubrics break down a final product into component parts, and each part is scored separately. The total score of a student's performance is the sum of all parts. Each component can be evaluated and provide teachers with specific information about strengths and weaknesses that can guide instructional choices to help students improve.

Whether holistic or analytic rubrics are used, several significant issues need to be addressed. Raters' understanding of the scoring task and their ability to score observed behaviors in consistent ways are essential when making judgments about student performance. Consistency within an individual evaluator is also important—this is, does the rater score the performance in a similar manner on more than one occasion? Also, how many raters are required for confidence in scores? How similar are raters' scores on the same performances? Are their ratings similar on different occasions? Differences in scores may also be related to the task at hand. For example, the passages students read aloud may influence their abilities to perform well.

Currently, the two most commonly-used rating rubrics are the NAEP and the MDFS. However, another rating instrument, CORFS, has also been developed recently. The rating method created and used by the National Assessment of Educational Progress (NAEP) is a holistic measure [14,34]. The Multidimensional Fluency Scale (MDFS) is an analytic approach that measures four dimensions of expressive oral reading [35]. The Comprehensive Oral Reading Fluency Scale (CORFS) uses two factors to measure prosodic reading analytically [24].

## 6.1. NAEP

This measure was developed for the 1992 NAEP assessment, the first time that fluency was assessed since NAEP began 25 years earlier. The prosody measure was developed by Gay Sue Pinnell, John J. Pikuiski, Karen K. Wixson, Jay R. Campbell, Phillip B. Gough, and Alexandra S. Beatty, who all served on the NAEP fluency committee [14]. Fluency has only been measured one additional time for fourth graders in 2002. In both cases, NAEP used the same instrument to measure rate, accuracy, and fluency, the term they used to describe prosody. These scholars designed this holistic measure that focuses primarily on phrasing, syntax, and expression: "In this study, fluency was considered a distinct attribute of oral reading separate from accuracy and rate. Fluency was defined in terms of phrasing, adherence to the author's syntax, and expressiveness, and was measured at one of four levels (1–4, with 4 being the measure of highest fluency) on NAEP's Oral Reading Fluency Scale." Those who scored in levels three and four were considered to be fluent, and those who scored in levels one and two were non-fluent (see Table 1). Although there was a new committee in 2002, they chose to use the exact same prosody measure. [34].

**Table 1.** National Assessment of Educational Progress (NAEP) Fluency Scale.

| Category | Level | Description |
|----------|-------|-------------|
| Fluent | 4 | Reads primarily in larger, meaningful phrase groups. Although some regressions, repetitions, and deviations from text may be present, these do not appear to detract from the overall structure of the story. Preservation of the author's syntax is consistent. Some or most of the story is read with expressive interpretation. |
| | 3 | Reads primarily in three- or four-word phrase groups. Some small groupings may be present. However, the majority of phrasing seems appropriate and preserves the syntax of the author. Little or no expressive interpretation is present. |
| Non-fluent | 2 | Reads primarily in two-word phrases with some three- or four-word groupings. Some word-by-word reading may be present. Word groupings may seem awkward and unrelated to larger context of sentence or passage. |
| | 1 | Reads primarily word-by-word. Occasional two-word or three-word phrases may occur—but these are infrequent and/or they do not preserve meaningful syntax. |

The fluency assessment method called for individual interviews with a sample of fourth graders during which they were recorded reading one page of text orally. Recordings were analyzed by trained raters who used the rubric that had been developed. In 2002, a sample of 1779 fourth-graders from the total of 140,000 students included in the NAEP reading assessment were interviewed. Findings revealed that 10% of all readers were rated in the highest, or fourth, level of expressive reading, and 51% of the sample were rated in third level. The overall mean score for the 2002 fourth grader sample was 2.64 out of a possible 4 points. Additionally, data showed that as rate, accuracy, and prosody increased, so did comprehension. These results demonstrated a strong relationship between fluency and comprehension: "Skilled readers not only recognize and read words quickly, but also deliver a smooth oral reading performance that reflects their understanding of the text they are reading." These authors report that two raters scored each oral reading with an intraclass correlation of 0.82. The number of rating occasions beyond one is unknown [34].

A number of studies have used NAEP to examine expressive oral reading. Morris et al. examined rate, accuracy and prosody of first graders reading short passages to determine which factors best predicted scores on fluency ratings. Using the NAEP fluency assessment measure, they found that rate and phrasing surfaced as the best predictors [36].

In a study of second graders, Tortorelli utilized a statewide assessment database of rate, accuracy, prosody, and comprehension. The prosody score was obtained using the NAEP measure. She compared results of those who read slowly to four other groups: those with generally high skills; those with high accuracy and low rate scores; those with low accuracy and high rate; and those with generally low scores. She found that those who struggled with accuracy also demonstrated difficulty with comprehension. Those who struggled with rate also demonstrated difficulty with prosody [37].

In summary, the NAEP measure shows variability in expressive oral reading with a 2002 mean fluency score of 2.64. When two raters are used, the instrument is found to be reliable (correlation of 0.82). In the 2002 NAEP assessment, nearly two-thirds of the fourth graders read at a fluent level. Phrasing, a key element of prosody, surfaced as a predictor of overall fluency. This scale also showed that those who struggle with word identification also demonstrated low prosody.

*6.2. MDFS*

The Multidimensional Fluency Scale (MDFS) has also been used to assess expressive oral reading. This measure was initially developed by Zutell and Rasinski [18], who were influenced and motivated to do so by the work of two groups of researchers, Allington and Brown [38], and Aulls [39]. Zutell and Rasinski noted that both groups of scholars had identified specific elements of prosody [18], and that Aulls had created a rough scale for observing stages of reading fluency (word-by-word reading and phrasing, and expression). However, Aulls did not include all of the same elements that Zutell and Rasinski had described in greater detail in their initial Multidimensional Fluency Scale [18]. Zutell and Rasinski were also influenced by the NAEP prosody rating scale that was being developed for use in 1992. The MDFS was "an elaboration of the fluency rubric used in the NAEP studies of oral reading [14,34] that reported significant correlations (predictive validity) between oral reading prosody and fourth-grade students' silent reading comprehension" [35]. Unlike the NAEP scale, the MDFS utilized an analytic scoring system, using four levels, from low to high prosody focusing on three domains of expression—phrasing, smoothness, and pacing. A separate score for each of the three traits of expressive reading was given.

The MDFS changed from the original three to four dimensions of prosody in 2003, when the additional aspect of expression and volume was given [40]. Table 2 shows the current MDFS that consists of a four-point scale for four specific dimensions of fluency—expression and volume, phrasing, smoothness, and pacing.

**Table 2.** Multidimensional Fluency Scale (MDFS).

| Dimension | 1 | 2 | 3 | 4 |
|---|---|---|---|---|
| Expression and Volume | Reads with little expression or enthusiasm in voice. Reads words as if simply to get them out. Little sense of trying to make text sound like natural language. Tends to read in a quiet voice. | Some expression. Begins to make text sound like natural language in some areas of the text, but not others. Focus remains largely on saying the words. Still reads in a quiet voice. | Sounds like natural language throughout the better part of the passage. Occasionally slips into expressionless reading. Voice volume is appropriate throughout the text. | Reads with good enthusiasm throughout the text. Sounds like natural language. The reader is able to vary expression and volume to match his/her interpretation of the passage. |
| Phrasing | Monotonic with little sense of phrase boundaries, frequent word-by-word reading. | Frequent two- and three-word phrases giving the impression of choppy reading; improper stress and intonation that fail to mark ends of sentences and clauses. | Mixture of run-ons and mid-sentence pauses for breath, and possibly some choppiness; reasonable stress/intonation. | Generally, well phrased, mostly in clause and sentence units, with adequate attention to expression. |
| Smoothness | Frequent extended pauses, hesitations, false starts, sound-outs, repetitions, and/or multiple attempts. | Several "rough" spots in text where extended pauses, hesitations, etc., are more frequent and disruptive. | Occasional breaks in smoothness caused by difficulties with specific words and/or structures. | Generally smooth reading with some breaks, but word and structure difficulties are resolved quickly, usually through self-correction. |
| Pacing | Slow and laborious. | Moderately slow. | Uneven mixture of fast and slow reading. | Consistently conversational. |

To use the MDFS, raters make judgments about individuals' prosodic reading in each of the four dimensions. The descriptions of the levels allow raters to make consistent decisions about readers' performances. However, with so many decisions required of raters, questions arise about the reliability and validity of the scores obtained using this instrument.

The reliability and validity of scores obtained using the MDFS have been established by various researchers. Zutell and Rasinski said that "initially teachers often feel insecure in making 'subjective' judgments; they are concerned about issue of reliability and validity" [18]. To alleviate these concerns, they conducted research to show that fluency ratings are strong predictors of results on standardized reading tests [41]. Further, they showed that with training, university teacher candidates could learn to apply to rubric accurately and consistently [42]. The training of raters is important to ensure reliability of scores, but questions remain about the number of raters, passages, and rating occasions that are required to obtain reliable scores. The combination of raters, passages and occasions also make the feasibility of using the MDFS an issue.

Moser, Sudweeks, Morrison, and Wilcox addressed these specific issues in a generalizability study of ratings of 36 fourth- graders' reading. For three days each week over a seven-week period, these students practiced fluent oral reading of passages from both genres. At the conclusion of that practice period, students read four passages—two narrative and two informational—to their teacher, the lead researcher in the study. All readings were recorded so that expressive oral reading could be assessed on different occasions. The 144 readings were evaluated using the MDFS by two trained raters on two separate rating occasions.

Results show the mean score for expression and volume was 3.11, phrasing was 3.25, smoothness was 3.12, and pace was 3.06, with an overall mean of 12.54 out of 16 [43].

Generalizability theory methods were used to evaluate the rating scores. These procedures provided a way to simultaneously estimate main effects and interaction effects through the analysis of mean ratings. Generalizability theory goes beyond the traditional ANOVA in that it can be used to estimate the relative percentage of measurement error from multiple facets. In this way, researchers were able to estimate the reliability of scores obtained using the MDFS. By using Generalizability theory, researchers also examined effects related to raters, rating occasions, and passages. Results were used to estimate the number of raters, rating occasions, and passages that are required to obtain reliable scores for expressive oral reading.

Results showed very high MDFS reliability scores, ranging from 0.92 to 0.98. Findings also showed that a minimum of two, and preferably three, equivalent passages, two raters, and one rating occasion are recommended to obtain reliable ratings. Like the research by Zutell and Rasinski, this study also demonstrated the value of training raters and encouraging them to collaborate during training sessions. In addition, this study showed the necessity of using multiple passages along with multiple raters. Most important, this study found that highly reliable expressive oral reading scores can be obtained using the MDFS and assures researchers and teachers that it can be used to measure expressive oral reading.

Smith and Paige were interested in examining the reliability of scores on prosodic reading that can be obtained using both the NAEP fluency scale and the MDFS. They sought to compare these two measures of prosody. Like Moser et al., they also used Generalizability theory. They trained four doctoral students to use both the MDFS and NAEP to rate children's oral reading. Results showed an average NAEP score of 2.54 out of 4 on the first occasion and 2.70 on the second. They also showed scores of 10.09 out of 16 and 10.76 on the two separate occasions using MDFS [44]. Children in first, second, and third grade orally read one grade-level, narrative passage from the Gray Oral Reading Test-5 [45]. All readings were digitally recorded so that ratings of prosody could be completed. The four raters judged the oral reading of 177 readers on two occasions.

These researchers measured the amount of variance contributed by differences in raters, rating occasions, and students. Reliability coefficients were very similar for the MDFS and the NAEP. Results showed high reliability scores for each of the three grade levels, ranging from 0.91 to 0.94 for both rating instruments. Results showed slightly higher reliability scores for the MDFS than the NAEP, but the two measures were highly correlated with no significant differences in scores obtained from the two instruments.

Although the MDFS and NAEP produced similar results, the MDFS was slightly more efficient than NAEP in regard to measurement design resources. To obtain desired results, the MDFS required only two raters, as opposed to three needed when using the NAEP instrument. MDFS provided a deeper analysis of the quality of reader fluency, due to the analytic nature of the MDFS and the holistic quality of the NAEP measure. The precision of information from the MDFS can better inform instruction. Training raters was essential to obtain reliable scores, regardless of which rating scale was employed.

The MDFS has been used in a number of studies that examined expressive oral reading. Dutch researchers examined the oral reading of 106 fourth graders to see what aspects are closely associated with reading comprehension. They used the MDFS to measure prosody. Regression analyses showed that prosody, not rate, was most closely linked to comprehension scores [46]. Similarly, Turkish researchers examined the oral reading of 132 fourth graders. Using the MDFS, they showed a strong relationship among attention, reading speed, and prosody [47].

Repeated readings have been suggested as a way to improve oral reading fluency. Guerin and Murphy used the MDFS in their study of struggling adolescent readers. Results showed that over a seven-week period of repeated reading, all aspects of fluency improved, leading to more strategic reading and improved comprehension [48].

In summary, the MDFS measure shows variability in expressive oral reading with mean fluency scores of 10.76 out of 16 in the Moser et al. study and 12.54 in Smith and Paige. This instrument had a reliability score of 0.92–0.98 for the Moser et al. study and 0.91–0.94 in Smith and Paige. These reliability scores assume two raters and one rating occasion. Other researchers using this scale have found that prosody better predicts comprehension than rate, even though the two have a strong relationship. Another result is that improved expressive reading can lead to greater reading comprehension.

### 6.3. CORFS

The Comprehensive Oral Reading Fluency Scale was developed in 2013 by researchers who had used automated means of assessing prosody [24]. They sought to create a spectrographically-grounded rating scale. As a result, the two features of prosody measured with this rubric were intonation (pitch) and pausing (see Table 3). There are four possible rating levels for both dimensions.

**Table 3.** Comprehensive Oral Reading Fluency Scale (CORFS).

| Automaticity (Circle Rating) | | Expression (Circle Ratings) | | | |
|---|---|---|---|---|---|
| Rating | WCPM | Intonation Rating | Appropriate Intonation | Pausing Rating | Natural Pausing |
| 8 | 137+ | 4 | • Makes noticeable pitch variations throughout to communicate meaning. • Makes appropriate and consistent end-of-sentence pitch changes. • One or two exceptions may exist. | 4 | • Pauses may be used to convey meaning. • Between-sentence pauses are short but natural. • Unexpected pauses occur less than once per sentence on average. |
| 6 | 107–136 | 3 | • Varies pitch appropriately and makes appropriate end-of-sentence pitch changes most of the time. • Some flatness may exist, but intonation effectively communicates meaning overall. | 3 | • May have brief unexpected pauses once or twice per sentence, but pauses seem to be used mainly to distinguish phrases and sentences. • Longer pauses are rare and only momentarily interrupt the flow of the text. |
| 4 | 78–106 | 2 | • Intonation is frequently flat or does not match the punctuation or meaning/ phrasing of the text. • Shows appropriate pitch variation on a few sentences but is flat or unnatural on many others. • Overall impression is that intonation does not effectively communicate meaning. | 2 | • Frequent pausing within sentences. • May also have some lengthy pausing between sentences. • May pause often between phrases or three- or four-word groupings. |
| 2 | 1–78 | 1 | • Reads with flat or other unnatural intonation throughout. • Does not mark sentence boundaries with distinct pitch changes, except occasionally. | 1 | • Reading is broken and effortful with numerous pauses throughout. • Reads primarily in groups of one or two words without pausing. |

Benjamin et al. thought that this instrument would be an advance over MDFS and NAEP because of its grounding in the more objective spectrographic research literature on reading prosody [24]. CORFS assesses the three components of fluency (i.e., rate, accuracy, and prosody) simultaneously by incorporating measures of all three components, capitalizing on the published grade-level reading rate norms by Hasbrouck and Tindal [49]. Benjamin et al. emphasize the understanding that fluency serves as a support for comprehension and that reading prosody is a critical component in this process [24].

In the development of CORFS, developers carried out two studies to evaluate its validity. In the first, three raters judged the expressive reading of 59 second graders. High degrees of correlation were found between these expression ratings and standardized assessments of reading comprehension and spectrographic measures of fluency. In the second study, 60 third graders read two texts and were evaluated using spectrographic measures. Then two raters independently evaluated the recordings using CORFS. Comparisons between the two prosody measures showed comparable results.

To our knowledge, this instrument has not been used by researchers outside those who developed it or who are involved in spectrographic research, although it has been cited by other researchers [50,51]. On other occasions the developers have suggested CORFS as a resource for other researchers [52,53].

In summary, the CORFS measure shows high correlations with spectrographic scores and traditional reading assessment scores. The mean CORFS score for total expression was 5.78 out of 8 and the interrater reliability coefficient was 0.99. Because it has not been used widely, it is not yet possible to discuss findings from studies using this instrument.

## 7. Discussion

Two general approaches have been used to evaluate expressive oral reading, automated assessment and human judgment. Bolaños et al. have completed a study using both [54]. They examined oral reading of first through sixth graders who read one of 20 passages written at their grade level in a one-minute timed reading. The 783 total recordings were evaluated for rate, accuracy, and expression using both automated and human scoring systems. The automated approach was developed by the authors who also trained teachers to calculate wcpm and then rate prosody using the NAEP scale [55].

Results showed high agreement between scores obtained using automated assessments and human ratings for both wcpm and prosody. The agreement between human and automated ratings of expressive oral reading were higher (90.93%) when judging between the fluent (levels 3 and 4) and non-fluent (levels 1 and 2) NAEP categories (76.05%). Nevertheless, the agreements were high enough that the authors conclude that we could settle for using automated ratings of all three aspects of fluency.

The concerns raised by other researchers are that digital measures may not accurately assess all dimensions of fluency [56]. For example, Smith and Paige reported higher reliability results for NAEP than those reported in the Bolaños study [44]. The MDFS, not used in the Bolaños study, has also been shown to yield highly reliable scores [43]. Automated tools are limited to measuring only pitch and pauses, leaving out other dimensions of prosody like smoothness and phrasing, which human raters may be better equipped to evaluate. Automated assessment can minimize qualitative relationships. For example, pitch and pausing contours may not always be correctly associated with the linguistic elements in the text. The same passage can be appropriately read with multiple prosodic patterns by proficient readers, while struggling readers may exhibit pauses and pitch differences that are not appropriate. There are times when words in a text can be grouped into more than one acceptable way and still preserve meaning. These groupings may not all be grammatically correct, and a spectrographic analysis generally would not accept such variations. A human rater can pay attention to why students are pausing as they read (e.g., to check what they've read, to decode the word they are currently reading, to anticipate an upcoming word, to take a break in their reading because they are tired). Spectrographic research can document these pauses, but only a human rater has the potential of understanding why.

On the flipside, use of human rating scales of prosody also has potential drawbacks. Whenever multiple raters are used, reliability and validity issues are drawn into question. Researchers have

shown that training of raters is essential to ensure inter- and intra-rater reliability. But that takes time and effort, which may not always be practical in a school setting.

Different rating scales have included different traits of prosody, showing disagreement about what constitutes expressive oral reading. Human raters make subjective judgments. Although such judgments can be tempered with multiple raters, passages, and rating occasions, educators may not have the luxury to go to such lengths. Human raters can make a holistic judgment using NAEP, but when using MDFS or CORFS they must make multiple decisions at the same time. It is difficult to make separate judgments for each dimension, and time consuming to listen to the recording multiple times to rate each dimension separately.

## 8. Conclusions

Recognizing the limitations of both types of assessment, perhaps the decision of which to use needs to be made by considering practical issues. Researchers may have access to the technology required for automated assessment, but most teachers will not. Those who can use this technology, may find that it saves time. But results may not inform instruction as effectively as one-on-one testing or listening to recordings. Teachers may be able to obtain the information they need about prosody without having to deal with multiple raters and rating occasions. Researchers, on the other hand, have to make careful and defensible choices about multiple raters, passages, and occasions to ensure the validity and reliability of their findings.

Whether teachers or researchers, one thing is clear—we have to measure all aspects of reading and not just decoding. If we are serious about improving children's comprehension, there is a place for assessment of expressive oral reading. If we take the time to teach and measure elements of prosody, we can make room for meaning—the primary purpose of reading.

**Author Contributions:** The two authors contributed equally. All authors have read and agreed to the published version of the manuscript.

**Funding:** This research received no external funding.

**Conflicts of Interest:** The authors declare no conflict of interest.

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
