# Peer review of "Assessing Expressive Oral Reading Fluency"

_education, doi:10.3390/educsci10030059_

Round 1

Reviewer 1 Report

This manuscript provides interesting historical background on how fluency has been understood in the context of competent reading. It also reviews assessments of fluency and how various measures deconstruct fluency.

The manuscript makes no mention of how relevant research studies were identified or what criteria were used for including a study in this review: for example what search terms were used, which data bases were explored, what types of research reviewed were included or excluded from the manuscript? Including such information would add to the scientific soundness of the manuscript.

Typo

Page 10, line 389: "...there is a place FOR assessment of expressive oral reading".  (Original text says "of".)

History

Page 1, lines 34 - 40: This paragraph talks about the fact that fluency was largely overlooked for "decades".  The last sentence of the paragraph however states "However, during that period attention that was paid to oral reading fluency increased." I would suggest rewording this sentence as it seems to contradict what has come before it in the paragraph. Perhaps omit it or use the same thought as an opening to the next paragraph.

Prosody

Page 2, lines 73 - 88 should be expanded as this paragraph is central to understanding why assessment of fluency is important. Identifying ways to accurately and easily assess the component aspects of fluency (each of which is mentioned in this paragraph) is a major goal of this manuscript.

NAEP, MDFS, and CORFS

Add a table to summarize the findings of the studies that used each of the rubrics NAEP, MDFS, and CORFS.

Discussion

The last paragraph of the manuscript comes down quite hard on DIBELS. Throughout the manuscript, fluency has been defined and a case has been made for specific assessments. If the author(s) want to include DIBELS in their discussion then a section should be added to the manuscript which reviews the research on that assessment tool. Without such a section, the manuscript would be more professional if this paragraph was reworked to leave out reference to "one-minute times tests" and to the DIBELS.

Author Response

Thank you for your careful review. It helped us with our revisions. We fixed the typo you indicated, as well as several others we found as we revised. We deleted the paragraph you suggested in your "History" comment. We like the idea of adding a table to summarize the findings for the three human rater assessment types. However, as we started going that direction, we realized that the data in the studies were not aligned in a way that permitted fair comparisons. In place of the table, we added a paragraph at the end of each of these three sections that briefly summarized the content of the articles we reviewed. We hope this is enough to allow readers to understand the main points. We did take out the reference to DIBELS and one-minute timed tests, rather than include an earlier section to review progress-monitoring assessment. That discussion would have detracted from our focus in the article. We feel like the changes we made strengthen the article, and we hope you agree. You can find our major revisions in red font throughout the article. Thanks again!

Reviewer 2 Report

The study offers a review of the scientific literature on expressive oral reading. Its contribution consists of offering an analysis and contextualized discussion of the most outstanding tools for the evaluation of this type of reading. In this sense, it would be necessary to explain what the criteria for bibliographical selection were.

It is suggested:

  1. Line 1. Replace the identification of the manuscript ("article") with "Review".
  2. Line 2. Specify the title (ambiguous).
  3. Include a section that details, with precision, the methodology used.
  4. Line 392. Include a section of conclusions clearly differentiated from the discussion section.

Author Response

Thank you for your careful review of the article. The suggestions you offered help a great deal as we revised the manuscript. You asked for us to describe the criteria we used to select articles for inclusion. We have done this at the end of section 3, Approaches to Prosody Assessment. We changed "Article" to "Review" on line 1. we did not change the title of the article. You wrote "specify the title (ambiguous)." We were uncertain about what you meant by this: do we need to change the title or set it apart from the rest of the text. It seems that the font size sets it apart physically, but the title seems to fit the requirements of our assigned topic for this themed issue of the journal. You could suggest an alternate title and pass along to the guest editors of this issue. We added a Methodology section as you suggested re-numbered the subsequent headings. We added a Conclusions section at the end of the article, as you suggested. We did not put these headings in red font, but you will find our major additions and revisions in red font. We feel like your suggestions made the article stronger, and we hope you feel the same. Thanks again!

Round 2

Reviewer 2 Report

Suggestions and recommendations have been addressed